# Sire Effects on Carcass of Beef-Cross-Dairy Cattle: A Case Study in New Zealand

**DOI:** 10.3390/ani11030636

**Published:** 2021-02-27

**Authors:** Natalia Martín, Nicola Schreurs, Stephen Morris, Nicolás López-Villalobos, Julie McDade, Rebecca Hickson

**Affiliations:** 1School of Agriculture and Environment, Massey University, Private Bag 11 222, Palmerston North 4442, New Zealand; N.M.Schreurs@massey.ac.nz (N.S.); S.T.Morris@massey.ac.nz (S.M.); N.Lopez-Villalobos@massey.ac.nz (N.L.-V.); R.Hickson@massey.ac.nz (R.H.); 2Greenlea Premier Meats Ltd., P.O. Box 87, Hamilton 3240, New Zealand; Julie@greenlea.co.nz

**Keywords:** beef-on-dairy, carcass fat, crossbreeding, dairy-beef, genetics, meat yield, progeny test

## Abstract

**Simple Summary:**

Cattle born in the dairy industry are a very important source of beef. This study evaluated the carcasses of cattle born to dairy cows and sired by a range of Angus and Hereford sires. Sire affected most carcass traits of their progeny, particularly size and fat traits. The heaviest sire had 46 kg greater carcass weight compared with the lightest sire, equivalent to NZ$266 greater value per progeny. Carcass fat traits (rib fat depth and marble scores) were the most variable among sires, indicating possibility of selection. Thus, using beef-breed sires chosen for greater carcass weight has the potential to increase the meat production of cattle born on dairy farms, while maintaining adequate fat levels and carcass quality to receive optimum payment.

**Abstract:**

There is interest in increasing the carcass value of surplus calves born in the dairy industry that are reared for beef production in New Zealand. This experiment evaluated the carcass of Angus and Hereford sires via progeny testing of beef-cross-dairy offspring grown on hill country pasture. Weight and carcass traits were analyzed from 1015 animals and 1000 carcasses of 73 sires. The mean of the progeny group means was 567 kg for live weight at slaughter, 277 kg for carcass weight, 48.9% for dressing-out, 240.3 cm for carcass length, 73.6 cm^2^ for eye muscle area, 7.4 mm for rib fat depth, 0.91 for marble score, 3.05 for fat color score, 3.01 for meat color score, and 5.62 for ultimate pH. Sire affected (*p* < 0.05) carcass size and fat traits, but not fat color, meat color, or ultimate pH (*p* > 0.05). There was a 46 kg increase in carcass weight between the best and worst sires tested. Carcass fat traits were the most variable among sires. The use of genetically superior beef-breed sires over dairy-breed cows has the potential to increase carcass weights from surplus calves born in the dairy industry, while maintaining adequate fat levels and carcass quality.

## 1. Introduction

An animal suited for beef production should grow quickly and produce a heavy carcass with an appropriate amount of subcutaneous fat which in turn needs to be of suitable eating quality for a particular market. In the New Zealand meat payment schedule, carcass weight and fat grade drive the payment to farmers [1]. The current system classifies carcasses according to maturity, sex, fat content, and muscling [2], where the “P” fat class grade typically achieves the highest price per kilogram of carcass weight and requires 3–10 mm of subcutaneous fat over the eye muscle at the twelfth rib. Within each grading class, an increase in carcass weight range will increase the value of the carcass. Thus, farmers are rewarded for producing heavy carcasses [3], with high saleable meat yields and within some constraints in terms of quality and appearance. 

Meat quality characteristics are those attributes of the beef product that determine the acceptability and value for the consumer [4], including visual aspects such as meat and fat color, as well as intramuscular fat in the muscle (commercially known as marbling) which is related to cooking and eating quality [5,6]. Many characteristics of meat, such as meat color, are influenced by ultimate pH [7,8]. The ultimate pH of beef produced in New Zealand under grazing conditions typically ranges between 5.5 and 5.8 [9,10,11,12]. Generally, beef with a pH range of 5.4 to 5.6 has the most desirable properties for appearance and eating quality, while ultimate pH above 6.0 is associated with undesirable changes in those meat quality characteristics [7,8,13,14,15]. 

The main commercial driver for using beef-breed cattle for beef production in New Zealand appears to be their ability to achieve the minimum of 3 mm of subcutaneous fat and greater saleable meat yield with a higher proportion of meat distributed in the high-value primal cuts, at a lighter carcass weight, and with a younger animal compared with dairy-bred cattle [1]. However, cattle born in the dairy industry contribute 66% of the New Zealand’s beef production on a per-head basis, and around 44% of calves reared in the beef industry were born on a dairy farm [16,17,18,19]. 

Crossbreeding is an effective tool for increasing performance and profitability through heterosis [20,21]. Crossbreeding using beef-breed sires over dairy-breed cows can increase income from sales of the beef-cross-dairy calves born on the dairy farm, because of the expected higher growth rates producing heavier animals with higher dressing-out or meat yield percentages compared with dairy-bred cattle [22,23]. Furthermore, there is a possibility to identify sires with good genetic merit for both calving ease and carcass weight [24]. Given that the terminal sire has a considerable direct genetic effect on the progeny, beef-breed sires with improved genetics, or better estimated breeding values (EBVs [25]), can be used to generate beef-cross-dairy calves better able to satisfy the requirements of the dairy, grower, and finisher farms, through to the meat processor.

Angus and Hereford are breeds widely used in New Zealand farming [26], and both breeds have selection programs to improve genetic merit for beef production. However, EBVs are calculated by BREEDPLAN [27] within each specific breed. Coleman [28] demonstrated that sires of Angus and Hereford breeds have a wide variation in performance for gestation length and birthweight, and that these traits are well predicted by their EBV. Martín, et al. [29] showed that growth trajectories differed among these sires selected based on their liveweight EBV for 600 days, and that differences between the lightest and heaviest sires increased from entry to the beef cattle farm at 4 months of age to finishing at 26 months of age. There is scarce information about the performance of beef-breed sires for carcass size, fat, and quality traits with beef-cross-dairy cattle in a pasture-based system, because beef-breed sire performance is usually measured through their pure beef-breed progeny. The hypothesis is that carcasses differ among sires used to produce beef-cross-dairy calves finished on pasture, and the selection of appropriate sires will increase the muscle and fat that contribute to saleable meat without disadvantage to meat quality traits. 

Therefore, the aim of this experiment was to evaluate carcass traits of a selection of Angus and Hereford sires via progeny testing of beef-cross-dairy offspring grown on hill country pasture. 

## 2. Materials and Methods

This experiment uses the same animals for which growth traits were previously reported [29]. This experiment was conducted at Limestone Downs, near Port Waikato, New Zealand (37°28′ S, 174°45′ E) with approval from the Massey University Animal Ethics Committee (15/65 and 18/50). Animals were processed commercially through Greenlea Premier Meats Ltd., Hamilton plant (37°48′ S 175°15′ E), in accordance with the standard New Zealand industry practice [30], with Halal certification.

### 2.1. Animals and Management

Angus-sired and Hereford-sired cattle born to dairy cows in spring 2016 (*n* = 531) and 2017 (*n* = 486) were included in the study. Calves were born to 2-year-old (primiparous) and mixed-aged (3+ years old, multiparous) dairy-breed cows, which were predominantly Holstein–Friesian or Holstein–Friesian–Jersey crossbred. Full details can be found in Martín, et al. [29]. Briefly, lactating mixed-aged cows were individually inseminated with semen from Angus and Hereford sires, which were selected on the basis of their EBVs so that, within each breed, a spread of birth weight, gestation length, and live weight at 600 days of age was achieved, except that birth weight EBV was restricted to the lighter 50% of the breed at the time of selection [28]. When similar sires were available, those with superior EBVs for intramuscular fat (IMF) and eye muscle area (EMA) were selected. The 15-month-old heifers were joined with either Angus or Hereford bulls by natural mating, and these sires were selected to be in the lightest 15% of breed for birth weight. 

The EBVs of each sire was obtained from the online databases of Angus and Hereford breed associations [27]. The data collected in this experiment was not included for the calculation of the BREEDPLAN EBVs for these sires. Mean and range of EBVs for carcass traits by breed of sire are presented in Table 1.

Calves were artificially reared on an allowance of 4 L of milk/head/day, and calf meal was offered during the transition from milk to pasture [28]. Calves were weaned at a mean live weight of 93.1 kg (SD 7.2) at a mean age of 81.8 days (SD 11.4). Once weaned, all calves were moved from the dairy platform to the sheep and beef hill country platform of the same farm. Male calves were castrated before 4 months of age. At 4 months of age (131.4 days old, SD 17.2), calves were allocated to 6 grazing herds based on live weight (light, intermediate, and heavy) and sex (female and male) and balanced for sire so that, where possible, all sires were represented in each grazing herd within year. In total, there were 12 grazing herds (2 years × 2 sexes × 3 liveweight groups), and animals remained in those herds throughout the experiment. All cattle were grazed on summer-dry hill country pasture on the coastal farm under commercial conditions [29].

### 2.2. Slaughter

The target liveweight for slaughter was 500 kg for heifers and 600 kg for steers. Each grazing herd was slaughtered as a complete group on the same day, when the mean live weight reached the slaughter target weight. The target was set so that most carcasses from steers and heifers achieved the “P” fat class grading (3–10 mm of subcutaneous fat). In addition, animals were visually assessed by a livestock buyer to ensure that most cattle reached “P” grade and a 2 (or 1) conformation score as a pre-slaughter requirement, which in the industry is known as the “finishing” condition. The buyer’s criteria were that cattle had a flat back, a second roll starting to appear on the tail and a full brisket [31]. Animals were processed commercially through Greenlea Premier Meats Ltd., in accordance with the standard New Zealand industry practice [30]. 

### 2.3. Measurements 

#### 2.3.1. Weights 

On the day of slaughter, live weight was measured on the farm through a weigh crate (cattle crush model Cattlemaster Titan, made by Te Pari Products Ltd., Oamaru, New Zealand; weigh scales model XR5000, Tru-Test, Auckland, New Zealand), within one hour after yarding from a nearby paddock and prior to transport. 

After slaughter, the bodies were dressed to commercial specifications and carcasses were halved through the midline. Hot carcass weight (kg) was obtained as the sum of the weight of each carcass half recorded prior to the carcasses going into the chiller. Dressing-out percentage was calculated as the hot carcass weight divided by the live weight measured on the farm (x100).

#### 2.3.2. In-Chiller Assessments for Carcass Traits

Carcasses were chilled (4 ± 1 °C) overnight and the following morning (approximately 8 to 15 hours after slaughter), the length of one side of the carcass was measured from the distal end of the tarsal bones to the mid-point of the cranial edge of the first rib [11]. The other side of the carcass was cut between the twelfth and thirteenth rib to expose the ribeye muscle (*M. longissimus thoracis*) for in-chiller assessment of marbling score, meat and fat color scores, rib fat thickness, and EMA. The EMA (cm^2^) was traced onto waterproof paper and subsequently measured using a Planimeter (Placom KP-90N, Tokyo, Japan). The rib fat thickness (mm) was measured with a ruler as the subcutaneous fat depth at the thirteenth rib over the deepest part of the ribeye muscle. 

Carcasses were assessed for marbling, and meat and fat color by an assessor qualified to AUS-MEAT standards [32]. Marbling and meat color were scored after the eye muscle had been exposed to air for 30 min. Possible marbling scores ranged from 0 (nil) to 9 (abundant) and assessed the amount of marbling present in the ribeye muscle. Meat color scores ranged from 1 (light) to 7 (dark) and assessed the color of the lean muscle. Fat color was assessed on the intermuscular fat lateral to the ribeye muscle and adjacent to the *M. iliocostalis*, with possible scores ranging from 0 (white) to 9 (yellow).

For the 2017 cohort only, the ultimate pH was measured by pH spear (Eutech Instruments, Singapore) on the chilled carcass, at three points from medial to lateral across the ribeye muscle at the quartering site. The mean of the three measurements was used for analysis.

### 2.4. Statistical Analysis

#### 2.4.1. Data Cleaning

Animals born to sires with a minimum of 5 progeny were included in the analysis (*n* = 1101 animals from 73 sires). Animals that went missing, were recorded with ill health, or were removed from their grazing herd for more than 2 months were excluded from analysis of slaughter, carcass, and meat quality traits (*n* = 53 in 2016 and *n* = 31 in 2017). Any carcass recorded with a defect was excluded from analysis of carcass traits due to potential trimming of the carcass prior to weighing (*n* = 12 in 2016 and *n* = 3 in 2017). In-chiller assessments of marbling, and meat and fat color scores for the heavy steers born in 2016 were not recorded (*n* = 101). The dataset consisted of 1017 animals and 1002 carcasses from 73 sires. 

#### 2.4.2. Contemporary Groups

For comparisons among sires, the contemporary group was defined as the group of animals grazing in the same herd and year (*n* = 12), which were progeny of dams of the same age (*n* = 2, 2-year-old and mixed-aged) and progeny of sires of the same breed (*n* = 2, Angus and Hereford). One contemporary group (light Angus-sired steers born to 2-year-old cows in 2016) had only 2 animals both from the same sire, and so these progeny were excluded from analysis, taking the final dataset to 1015 animals and 1000 carcasses from 73 sires. The remaining contemporary groups (*n* = 47) had between 6 and 43 animals, with an age range of 11 to 65 days between the youngest and oldest animal in each group. 

#### 2.4.3. Statistical Models 

Linear mixed models (SAS 9.4, SAS Institute Inc., Cary, NC, USA) were used to obtain the least-squares means of the progeny groups for weight and carcass traits. The models included the fixed effects of sire within breed, and the random effect of contemporary group (*n* = 47). Models for weight traits (live weight, carcass weight, and dressing-out) also included the animal’s age deviation from the median age of its contemporary group as a covariate. Models for carcass size and fat traits (carcass length, EMA, rib fat depth, and marble score) also included carcass weight as a covariate. 

A mean and standard deviation of the least-squares means of the progeny groups were calculated for each trait, with equal weighting per sire regardless of number of progeny. A coefficient of variation (CV) was calculated with the standard deviation and the mean of the least-squares means of the progeny groups. The distribution of the least-squares means of the progeny groups were graphed with boxplots.

## 3. Results

The target slaughter liveweight and finishing condition set for these beef-cross-dairy cattle were achieved. Heifers (*n* = 495) were slaughtered at a mean age of 27 months (SD 2) and 520 kg live weight (SD 38), whilst steers (*n* = 522) were slaughtered at a mean age of 29 months (SD 1) and 614 kg live weight (SD 42). Most carcasses (97%) were graded as “P” fat class (3–10 mm of subcutaneous fat) and only 28 carcasses (3%) were graded “L” (less than 3 mm of fat). All carcasses received a conformation score of 2 and had low marble scores between 0 and 3 (from a total possible range 0–9). Carcasses had meat color scores between 1 and 6 (from a total possible range 1–7) with 87% of carcasses scoring 3 or less. Fat color scores ranged from 1 to 5 (total possible range 0–9), with 78% of carcasses scoring 3 or less. The mean ultimate pH was 5.62 (SD 0.14), with 1.9% (*n* = 9/486) of carcasses with pH over 6.0.

The total number of progeny and the mean of the least-squares means of the progeny groups for weight and carcass traits are presented in Table 2. Least-squares means of the progeny groups for age at slaughter had a 103-day range (804–907 days of age) and was different among sires (*p* < 0.05). Breed of sire had no effect on age at slaughter (*p* > 0.05).

The distribution of the least-squares means of the progeny groups for weights at slaughter are presented in Figure 1. All weight traits differed among sires (*p* < 0.05) after adjustment for the age deviation of each animal within its contemporary group (covariate effect *p >* 0.05). Live weight pre-slaughter ranged from 534 to 617 kg (*p* < 0.05), carcass weight from 258 to 304 kg (*p* < 0.05), and dressing-out from 47.4 to 50.3% (*p* < 0.05). The breed of sire had no effect on any of these traits (*p* > 0.05).

The distribution of the least-squares means of the progeny groups for carcass size and fat traits are presented in Figure 2, which differed among sires (*p* < 0.05) after adjustment for carcass weight (covariate effect *p* < 0.05 for carcass length, EMA and rib fat depth, but *p* > 0.05 for marble scores). Carcass length ranged from to 234.5 to 245.2 cm (*p* < 0.05), EMA from 65.5 to 82.9 cm^2^ (*p* < 0.05), rib fat depth from 4.3 to 11.3 mm (*p* < 0.05), and marble scores from 0.21 to 1.58 (*p* < 0.05). The breed of sire had no effect on EMA or rib fat depth (*p* > 0.05), but carcasses of Hereford-sired cattle were 2.0 cm longer (1% CV, *p* < 0.05) and had marble scores 0.21 lower (on a scale of 0–9, 16% CV, *p* < 0.05) than Angus-sired cattle, as shown in the distribution of progeny mean for carcass length and marble score by sire (Figure 2a,d).

The distribution of the least-squares means of the progeny groups for carcass quality traits are presented in Figure 3. Fat color scores (range 2.48–3.94), meat color scores (range 2.63–3.57), and ultimate pH (range 5.57 to 5.73) were similar among sires (*p* > 0.05). The breed of sire had no effect on ultimate pH or fat color scores (*p* > 0.05), but carcasses from Angus-sired cattle had meat color scores that were 0.09 greater (on a scale of 1–7, 2% CV, *p* < 0.05) compared with Hereford-sired cattle, and this is shown in the distribution of progeny mean for meat color scores by sire (Figure 3b).

## 4. Discussion

The carcass weights obtained in this study were consistent with the range of carcass weights for New Zealand in 2019: 242 for heifers and 313 kg/head for steers [33]. However, slaughter and carcass weights differed among sires, with the potential to increase carcass weight by 46 kg through selecting the best versus the worst sire tested. With an average price of NZ$5.75/kg of carcass (average price paid by Greenlea Premier Meats to the 2017 steer cohort), the 46-kg difference in carcass weight represented NZ$266 difference in carcass value between progeny of the top and bottom sires in this study. In addition, the difference in carcass weight was achieved through selection of sires with a spread in live weight at 600 days of age [28], rather than the carcass weight itself, and therefore greater differences could be obtained by selecting specifically for carcass weight.

All dressing-out percentages in this experiment were within typical values for beef cattle in New Zealand of around 50% [34] but there were differences among sires. The sires with dressing-out greater than 50% had progeny with average live weights at slaughter (between 3 lighter to 0.5 kg heavier) but above average carcass weights (6 to 7 kg heavier), supporting previous evidence that dressing-out percentages increases with increasing carcass weight [4]. Consequently, selecting sires with heavier carcass weight will also improve dressing-out percentages and will result in a greater economic return to the farm. 

Most carcass size and fat traits were dependent on carcass weight, which was consistent with previous studies [4,35,36,37]. When progeny were compared at the same carcass weight, carcass length, EMA, rib fat depth, and marble scores differed among sires. Rib fat depth had a 19% CV among sires, with no sires being below 3 mm and only 2 above 10 mm. The range 3–10 mm of subcutaneous fat or “P” fat class grade [2] is typically the best paid grade in the New Zealand market. Within this range, carcasses have an adequate amount of fat for meat eating quality and contribution to saleable meat yield without the requirement for trimming [4,32,34]. Only 3% of carcasses were classified as “L” fat class grade (subcutaneous fat below 3 mm). These results indicate that adequate amounts of rib fat depth can be achieved with beef-cross-dairy progeny when grazed on pasture. In addition, it demonstrates that the cattle in this experiment were processed at similar carcass weights and conditions as most cattle on commercial farms, so the results are relevant to what could be expected in the New Zealand beef cattle industry.

All the beef-cross-dairy progeny of the sires included in this study had low marble scores (0 to 3 on a scale of 0–9), which is supported by previous studies with cattle finished on mixed pastures and slaughtered at 24–28 months of age in New Zealand [9,10,11,38]. Adjusted progeny average values recorded within a recent Beef Progeny Test in New Zealand (including Angus, Hereford, Stabilizer, Simmental, and Charolais breeds) reported values from 2.4 to 4.3% IMF [38]. Other studies in New Zealand have shown mean values in the range of 3.0–3.9% IMF for Angus and Hereford–Angus and 2.9% IMF for Hereford–Friesian steers [9,10,11]. These IMF values are equivalent to marble scores between 1 to 2 in the AUS-MEAT marbling reference standards [32,39], and support the current findings in this study.

Even with overall low marble scores, there was large variation between sires (35% CV) after adjustment to the same carcass weight. There were 14 sires with marble scores greater than 1.24, of which 4 were Hereford and 10 were Angus. The fact that there were more Angus than Hereford sires in the higher range of marble scores in this experiment can be explained by the emphasis on marbling as part of the AngusPure brand in New Zealand. The New Zealand Angus breeders have been selecting animals for higher marbling, specifically through the AngusPure Index, “targeting the production of grass finished steers at 525 kg live weight (280 kg carcass weight and 10 mm rib fat depth) at 18 months of age with a significant premium paid for marbling” [40]. To qualify for AngusPure premiums, heifers and steers require a minimum marbling score of 2 [41]. Thus, greater marble scores can be achieved in beef-cross-dairy animals by selecting sires that have been included in this type breeding index. 

There were sire differences for EMA and carcass length (after adjustment to the same carcass weight), although these were small. The low variability for EMA among sires (5% CV) indicated a low potential to improve retail beef yield by selection of sires with greater EMA [42,43,44]. The variation was only 1% CV for carcass length (or 10.7 cm difference between shortest and longest sire). Carcass length is an indirect measure of animal frame and size, as shorter animals have shorter carcasses [11]. Smaller-framed animals are typically early maturing compared with larger-framed animals, and tend to be younger when they move into a fattening phase of their growth [35,45]. This could be an advantage in pasture-based systems.

There were no differences due to sire for ultimate pH, in agreement with the low heritability estimates (h^2^ = 0.02–0.10) found for ultimate pH in beef [46,47]. Ultimate pH was found to be within the normal pH range for beef, with the exception of 1.9% of animals with a pH greater than 6.0. These cattle did not belong to a specific sire or breed. This is supported by earlier studies that showed that environment effects such as handling, transportation, and pre- and post-slaughter conditions, rather than genetic effects, contribute to muscle ultimate pH post-mortem [48,49,50]. 

Meat and fat color scores were not affected by sire, in agreement with their low heritability (h^2^ = 0.05–0.25 [5,42,51]). No carcasses were downgraded or classified as being too yellow. Fat can be yellow from animals grown on a pasture-based diet, because the yellow pigmentation comes from the accumulation of carotenoids from green forage in the fat [52,53]. This is especially true for those animals with Jersey parentage [53,54], and is negatively regarded in many countries, and thus can be penalized at the time of carcass grading [2].

One limitation of this experiment was the scarce information on the dams, and so maternal breed was not accounted for. Dams were predominantly Holstein–Friesian or Holstein–Friesian–Jersey crossbred. A greater proportion of Holstein–Friesian genetics would produce animals with heavier carcasses and darker meat color, while a greater proportion of Jersey genetics would produce lighter carcasses, higher marbling, and overall yellower fat, with greater variability in color [1,4,55,56]. Nevertheless, it is unlikely that there would be a bias favoring particular sires in the data from this study because sires were rotationally allocated to mating days and randomly allocated to cows in estrus on each mating day, and cows had similar live weights and milk production regardless of the sire they were bred with [28].

## 5. Conclusions

There were differences among beef-breed sires for carcass size and fat traits, but little variation in carcass quality of beef-cross-dairy cattle. There was a 46-kg increase in carcass weight between the best and worst sires tested, and thus selecting sires with heavier carcass weight will result in a greater economic return to the farm. Carcass fat traits (rib fat depth and marble scores) were the most variable among sires, indicating that greater fat levels could be achieved in beef-cross-dairy animals by selecting beef-breed sires with higher IMF or rib fat depth genetics. Therefore, the use of genetically superior beef-breed sires over dairy-breed cows has the potential to increase carcass weights from surplus calves born in the dairy industry, while maintaining adequate fat levels and carcass quality to receive optimum payment.

## Figures and Tables

**Figure 1 animals-11-00636-f001:**
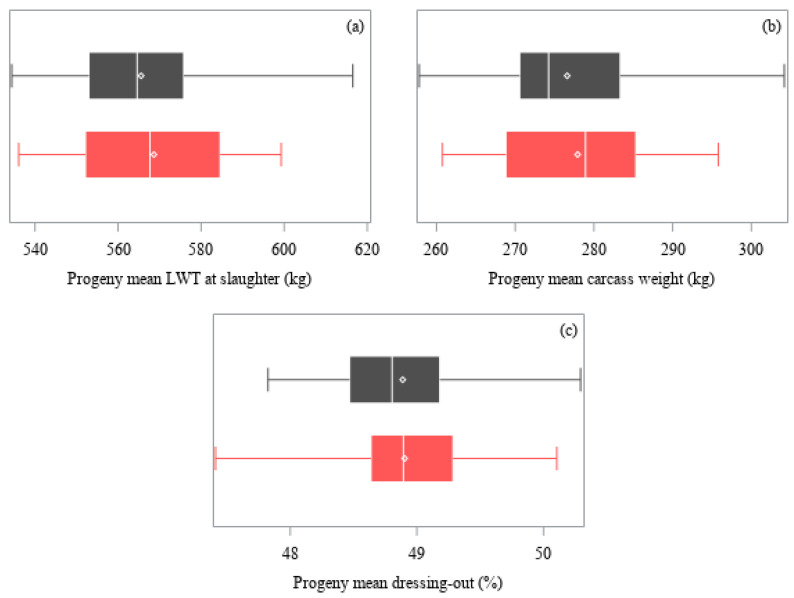
Distribution of least-squares means of the progeny groups of sires (Angus, ■ gray; Hereford, ■ red) for: (**a**) Live weight (LWT) at slaughter, (**b**) carcass weight, and (**c**) dressing-out percentage. Weight traits were adjusted by age deviation of the progeny within contemporary group. Each box represents the interquartile range (twenty-fifth to seventy-fifth percentiles), with the median value indicated by a line and the mean value indicated by a marker (**◇**). Whiskers extend to the minimum and maximum values.

**Figure 2 animals-11-00636-f002:**
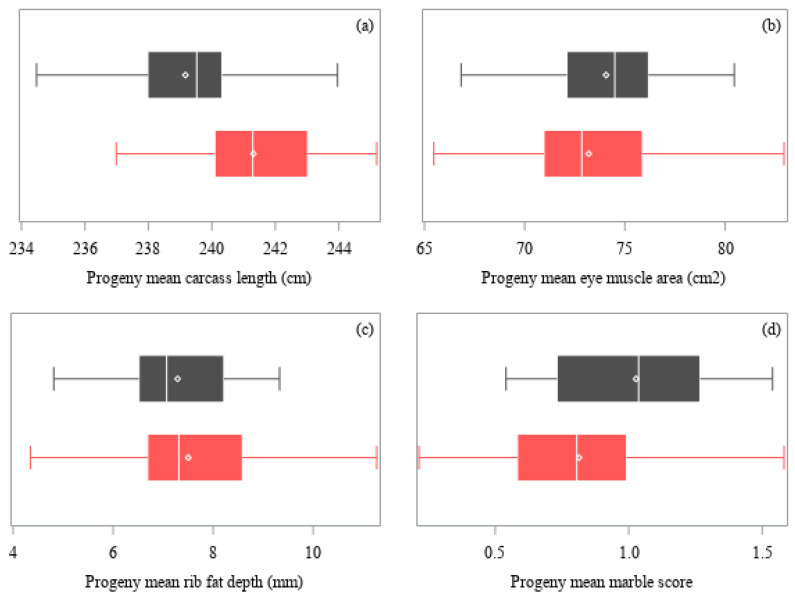
Distribution of least-squares means of the progeny groups of sires (Angus, ■ gray; Hereford, ■ red) for: (**a**) Carcass length, (**b**) eye muscle area, (**c**) rib fat depth, and (**d**) marble score. Carcass traits were adjusted by carcass weight. Each box represents the interquartile range (twenty-fifth to seventy-fifth percentiles), with the median value indicated by a line and the mean value indicated by a marker (**◇**). Whiskers extend to the minimum and maximum values.

**Figure 3 animals-11-00636-f003:**
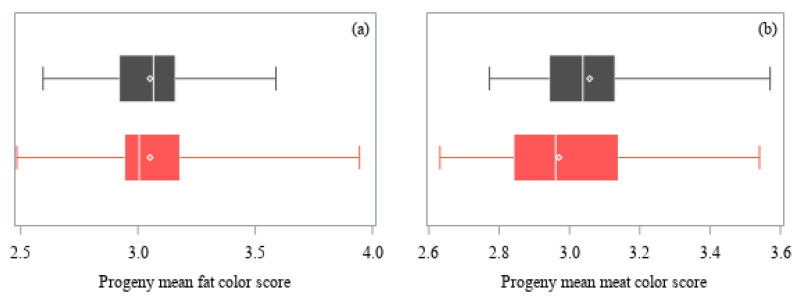
Distribution of least-squares means of the progeny groups of sires (Angus, ■ gray; Hereford, ■ red) for: (**a**) Fat color score, (**b**) meat color score, and (**c**) ultimate pH. Each box represents the interquartile range (twenty-fifth to seventy-fifth percentiles), with the median value indicated by a line and the mean value indicated by a marker (**◇**). Whiskers extend to the minimum and maximum values.

**Table 1 animals-11-00636-t001:** Estimated breeding values (EBV; mean ± SD) for carcass traits, for 37 Angus and 40 Hereford sires [27].

Trait	Angus			Hereford		
	*n*	EBV	EBV Range	*n*	EBV	EBV Range
Carcass weight (kg)	37	49 ± 16	(18 to 80)	40	53 ± 14	(25 to 84)
Eye muscle area (cm^2^)	37	4.6 ± 2.5	(−2.2 to 9.7)	40	2.9 ± 1.9	(0.3 to 8.0)
Rib fat (mm)	37	0.9 ± 1.8	(−2.0 to 6.1)	40	0.8 ± 0.9	(−1.8 to 2.7)
Intramuscular fat (%)	37	1.3 ± 1.4	(−2.1 to 4.4)	40	0.4 ± 0.7	(−1.0 to 2.0)

*n*: Number of sires used at mating; final number of sires included for data analysis: 34 Angus and 39 Hereford.

**Table 2 animals-11-00636-t002:** Number of progeny, mean (± SD) of the least-squares means of the progeny groups, coefficient of variation (CV%), and *p*-value for the effect of sire on weight and carcass traits (live weight, carcass weight, dressing-out, carcass length, eye muscle area, rib fat depth, marble score, fat color score, meat color score, and ultimate pH), for 73 sires.

Trait	*n*	Mean ± SD	CV%	Sire Effect *p-*Value
Age (days)	1015	854.6 ± 21.8	3%	<0.001
Live weight (kg) ^1^	1015	567.2 ± 17.9	3%	<0.001
Carcass weight (kg) ^1^	1000	277.3 ± 9.3	3%	<0.001
Dressing-out (%) ^1^	1000	48.9 ± 0.6	1%	<0.001
Carcass length (cm) ^2^	999	240.3 ± 2.3	1%	<0.001
Eye muscle area (cm^2^) ^2^	995	73.6 ± 3.5	5%	<0.001
Rib fat depth (mm) ^2^	994	7.4 ± 1.4	19%	<0.001
Marble score ^2^	902	0.91 ± 0.32	35%	<0.001
Fat color score	914	3.05 ± 0.25	8%	0.058
Meat color score	914	3.01 ± 0.19	6%	0.212
Ultimate pH ^3^	486	5.62 ± 0.03	1%	0.961

*n*: Number of progeny for all sires included. ^1^ Traits adjusted by age deviation of the progeny within contemporary group. ^2^ Traits adjusted by carcass weight. ^3^ Ultimate pH included data of 45 sires (second cohort).

## Data Availability

Data are available on request.

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
