# Peer review of "Sire Effects on Carcass of Beef-Cross-Dairy Cattle: A Case Study in New Zealand"

_animals, 2021, doi:10.3390/ani11030636_

Round 1
Reviewer 1 Report
The manuscript “Sire Effects on Carcass of Beef-Cross-Dairy Cattle: A Case Study in New Zealand” is a continuation of the previous paper on “Sire Effects on post-weaning growth of Beef-Cross-Dairy Cattle: A Case Study in New Zealand”. This mean that some informations, repeated in both papers, are written in the same way.
The topic is not new in animal research but is ever interesting and the paper is useful and contributes to existing knowledge.
The authors have to improve mainly the presentation of results which are sometimes repeated. For example when it comes to repeating numbers already easily readable on graphs.
For example, line 230 (total spread of 82 kg);
Line 231 dressing-out from 47.4 to 50.3% (2.9%). We all know the difference is 2.9%, I’d suggest the Authors to leave only the significance level in brackets.
About Discussion
Line 273: NZD$5.75 per kilo, I'd suggest to use NZ $ 5.75/kg
Lines 328-329 The Authors report a maximum heritability value (0.06-0.27), but the reference 46 referred to lambs (h2=0.18-0.27). I suggest to specify.
Reviewer 2 Report
This is a well written manuscript presenting a very well designed study. However, the aims of the study and thus also the conlcusion should be more elaborated in my opinion. As it reads, this seems very descriptive and it does not become readily obvious, what major question was answered. Beyond that I only have one question regrading the statistical model:
ll 194 ff: So, age was treated as a linear covariate? Did the authors consider other functions (quadratic e.g.)? Overall, I think, the model should be described better.
Reviewer 3 Report
The article “Sire Effects on Carcass of Beef-Cross-Dairy Cattle: A Case 2 Study in New Zealand” requires some corrections:
- In lines 37-38 you have written about the system of carcasses’ qualification – does it comply with EUROP system or what rating values it has.
- In line 88 I saw an information, that in the experiment were used animals that were rated earlier – this text is incomprehensible.
- - Line 100 includes indicator EBV – what is that indicator and how is it counted? The consequence of this lin eis line 109, in which there is mentioned about BreedPlan EBV. It is not clearly explained.
- Line 165 – information about measuring pH - at what time intervals were the pH measurements taken after slaughter?
- Line 344- what contribution of Jersey genes was present in the maternal material?
- Line 347 - What exactly caused the yellow color of the fat. Most cows are grazed on pasture and the fat is white.
- Line from 352 to 359 - Conclusions need to be redrafted as nothing comes out concrete, new and practical.
- Line 353 – it is obvious
- Line 354 - selection of stud dogs for what?
- Line 355 - there were no economic indicators reported
- Line 356 - what really influences marbling: environment or genetics?
- Line 357 - it is obvious that hybrids increase body weight and gains because there is a heterosis effect, i.e. hybrid exuberance
- Line 358 - application incomprehensible
Recommendations:
- The English language is incomprehensible in many places, so it should be reviewed by a native speaker or translated correctly
- The study should demonstrate specifically: the influence of the bull (exact characteristics of the bulls) or the environment (which environmental factors played a role here)? Is there an interaction of the genotype with the environment?
- What factor influenced the yellow fat?
- The discussion should be more specific and related to the features studied.
- -Conclusions should be precise according to the topic and purpose of the research and be proven at work
- Most of the literature (24 items) is more than 10 years old.
Reviewer 4 Report
L31: In alphabetical order.
L266-267: Delete this. This is aim, not discussion.
